# Dimethyl Fumarate Treatment Reduces the Amount but Not the Avidity of the Epstein–Barr Virus Capsid-Antigen-Specific Antibody Response in Multiple Sclerosis: A Pilot Study

**DOI:** 10.3390/ijms24021500

**Published:** 2023-01-12

**Authors:** Massimiliano Castellazzi, Caterina Ferri, Alice Piola, Samantha Permunian, Gaia Buscemi, Michele Laudisi, Eleonora Baldi, Maura Pugliatti

**Affiliations:** 1Department of Neurosciences and Rehabilitation, University of Ferrara, 44121 Ferrara, Italy; 2Interdepartmental Research Center for the Study of Multiple Sclerosis and Inflammatory and Degenerative Diseases of the Nervous System, University of Ferrara, 44121 Ferrara, Italy; 3Neurology Unit, “S. Anna” University Hospital, 44124 Ferrara, Italy

**Keywords:** multiple sclerosis, Epstein–Barr virus, dimethyl fumarate, antibody, avidity

## Abstract

(1) Multiple sclerosis (MS) is a chronic inflammatory disease of autoimmune origin. The Epstein–Barr virus (EBV) is associated with the onset of MS, as almost all patients have high levels of EBV-specific antibodies as a result of a previous infection. We evaluated longitudinally the effects of dimethyl fumarate (DMF), a first-line treatment of MS, on the quantity and quality of EBV-specific IgG in MS patients. (2) Serum samples from 17 MS patients receiving DMF were taken before therapy (T0) and after 1 week (T1) and 1 (T2), 3 (T3) and 6 (T4) months of treatment. Anti-EBV nuclear antigen (EBNA)-1 and capsid antigen (CA) IgG levels and anti-CA IgG avidity were measured in all samples. (3) Serum levels of anti-CA IgG were lower at T1 (*p* = 0.0341), T2 (*p* = 0.0034), T3 (*p* < 0.0001) and T4 (*p* = 0.0023) than T0. These differences were partially confirmed also in anti-EBNA-1 IgG levels (T3 vs. T0, *p* = 0.0034). All patients had high-avidity anti-CA IgG at T0, and no changes were observed during therapy. (4): DMF can reduce the amount but not the avidity of the anti-EBV humoral immune response in MS patients from the very early stages of treatment.

## 1. Introduction

Multiple sclerosis (MS) is an autoimmune chronic inflammatory disease of the central nervous system (CNS) of unclear aetiology that is characterized by demyelination and axonal loss [1]. MS usually affects young adults, and women more frequently than men, and is the leading non-traumatic cause of neurological disability in young people [2,3]. Clinically, MS is depicted by attacks or exacerbations, called relapses, which show dissemination in space and time [4]. MS is considered a multifactorial disease in which the combination of exposure to an environmental factor, such as an infectious agent, and a genetic predisposition can drive the development of the disease [5]. The main suspect in this setting is represented by the Epstein–Barr virus (EBV), a human γ-herpesvirus that can infect, activate and latently persist in B-lymphocytes for life [6]. For years, an EBV infection has been considered the potential trigger for the autoimmune process that sustains MS pathogenesis [7]. The seroprevalence of anti-EBV antibodies has been found to be higher in MS patients than in the healthy population [8], and this humoral response consisted mainly of anti-viral capsid antigen (CA) IgG and anti-Epstein–Barr nuclear antigen 1 (EBNA-1) IgG [9]. Elevated serum levels of anti-EBV IgG were associated with (i) increased risk of developing MS, (ii) MS disease activity, (iii) the conversion from clinically isolated syndrome (CIS) to definite MS and (iv) grey matter atrophy [10,11,12,13]. Despite the association between EBV and MS also being supported by the demonstration of the virus’s presence within MS demyelinated lesions [14], the strongest association still derives from seroepidemiological studies as recently highlighted in a longitudinal analysis of a cohort of more than 10 million people active in the US military over a 20-year period [15].

Although MS is a chronic disease with no definite cure, some disease-modifying treatments (DMT) can improve the course of the disease. Among these treatments are (i) injectable DMTs: interferon-beta (IFNβ) and glatiramer acetate; (ii) oral DMTs: dimethyl fumarate (DMF), diroximel fumarate, teriflunomide, sphingosine 1-phosphate receptor modulators and cladribine; (iii) parenteral administrated monoclonal antibodies: natalizumab, ocrelizumab, ofatumumab and alemtuzumab [2]. In a recent article, Persson Berg and colleagues demonstrated that a second-line treatment such as natalizumab was able to induce a moderate decrease in serum EBV-specific IgG levels, while the previous treatment with IFNβ kept them almost stable [16].

Serologic tests of anti-EBV antibodies are frequently used to define the status of the infection. It is possible to distinguish acute from past infection using only three parameters: (i) IgG antibodies specific to CA, (ii) IgG antibodies specific to EBNA-1 and (iii) the avidity of anti-CA IgG [17,18]. The presence of anti-CA IgG without anti-EBNA-1 IgG indicates an acute infection, while the presence of anti-CA IgG and anti-EBNA-1 IgG is typical of a previous infection [18]. However, serological results can sometimes be difficult to interpret, as anti-CA IgG can be present without anti-EBNA-1 IgG in the case of a previous infection, or both parameters can be detected simultaneously during reactivation. An isolated anti-EBNA-1 IgG profile may also raise some doubts. To correctly interpret these patterns, the avidity of CA-specific IgG must be determined [19]. In particular, the presence of low-avidity antibodies indicates a recent infection (acute phase), while the presence of high-avidity antibodies is indicative of a past infection [19].

Starting from the assumption that high levels of EBV-specific antibodies are a biological characteristic of MS patients, and from the fact that these antibodies can be affected by DMT, this study aimed to evaluate longitudinally the effects of a first-line oral treatment on the amount and avidity of EBV-specific humoral immune response in a cohort of MS patients.

## 2. Results

### 2.1. Patient Characteristics

The demographic and clinical characteristics of 17 relapsing–remitting MS (RRMS) patients receiving DMF are listed in Table 1. The median age at entry was 40 years, and the F:M ratio was 2.3. The median disease duration at the time of enrolment was 29 months. The disease severity, expressed with the expanded disability status scale (EDSS) [20], ranged from 1 to 4, with a median of 1.5. During the study’s observation period, no patients experienced clinical relapses and no increases in disability were recorded on neurological examination.

### 2.2. Serum Levels of Anti-Epstein–Barr Virus Antibodies

All patients resulted seropositive for both anti-EBNA-1 and anti-CA IgG. Accordingly, serum levels of anti-EBNA-1 and anti-CA IgG, expressed as OD values, were determined in all samples. Median serum levels of anti-EBNA-1 IgG were different between various time points (Friedman test, *p* = 0.0066) (Table 2). Dunn’s post hoc test was then used for multiple comparisons of each time point vs. every other time point, for a total of ten comparisons. The median OD value at T3 was significantly lower than T0 (*p* = 0.0034). No further statistical significance emerged when comparing the median OD values at different time points. Median serum levels of anti-CA IgG resulted differences between the time points (Friedman test, *p* < 0.0001). In addition, in this case Dunn’s post hoc test for multiple comparisons was used to compare each time point vs. every other time point for a total of ten comparisons. Median OD values at T1 (*p* = 0.0341), T2 (*p* = 0.0034), T3 (*p* < 0.0001) and T4 (*p* = 0.0023) were significantly lower than T0. No further statistical significance was found comparing the median OD values at different time points.

Although no patient reported an increase in disability during the study period, the possible correlation between serum anti-EBNA-1 and anti-CA IgG levels with the EDSS scale was investigated. Our data did not show any statistical significance at T0 (Spearman’s test *r*: *r* = −0.4044 and *p* = 0.1080 for anti-EBNA-1 IgG; *r* = −0.2546 and *p* = 0.3210 for anti-CA IgG) or in any of the subsequent time points.

### 2.3. Changes in Serum Levels of Anti-Epstein–Barr Virus Antibodies

Changes in anti-EBNA-1 IgG levels between the paired samples collected before (T0) and during DMF treatment at different time points (T1, T2, T3 and T4) were analysed as delta (∆)OD. As reported in Figure 1A, a reduction in anti-EBNA-1 IgG levels was found in 88.2% of samples at T1, 82.4% at T2, 88.2% at T3 and 58.8% at T4. Compared to T0, ∆OD values were statistically reduced in T1 (*p* = 0.0005), T2 (*p* = 0.0017) and T3 (*p* = 0.0004) (Wilcoxon Signed Rank Test). Changes in anti-CA IgG were reported in Figure 1B. Negative ∆OD values were found in 88.2% of samples at T1, 100% at T2, 94.1% at T3 and 82.4% at T4. Referring to T0, ∆OD values were statistically reduced at all time points: T1 (*p* = 0.0002), T2 (*p* < 0.0001), T3 (*p* < 0.0001) and T4 (*p* = 0.0209) (Wilcoxon Signed Rank Test).

### 2.4. Anti-Epstein–Barr Virus Antibody Avidity

The avidity of anti-CA IgG was determined in all patients at the time points T0, T2, T3 and T4. We arbitrarily chose to consider the T1, corresponding to only 7 days, a negligible time point for the evaluation of the antibody avidity. The qualitative analysis highlighted that all patients had high-avidity antibodies at all time points as demonstrated by relative avidity index (RAI) values higher than 60% (Figure 2A). Quantitative analysis showed that median RAI values were not different among different time points (Friedman test, *p* = 0.9299) (Figure 2A). In particular, median anti-CA IgG avidity did not change between pre-treatment (T0) and the following time points: T2, T3 and T4 (Dunn’s post hoc test for multiple comparisons, all *p* > 0.9999). Even comparing the median OD values at different time points, no statistically significant differences emerged.

### 2.5. Changes in Anti-Epstein–Barr Virus Antibody Avidity

Changes in anti-CA IgG avidity between the paired samples collected during DMF treatment at different time points were also analysed as ∆RAI. As reported in Figure 2B, negative ∆RAI values, suggestive of a reduction in anti-CA IgG avidity, were found in 58.8% of samples at T2, 47.1% at T3 and 47.1% at T4. Compared to T0, median ∆RAI values resulted similarly at all time points without any statistically significant difference (Wilcoxon Signed Rank Test).

## 3. Discussion

In this study, we demonstrated for the first time that DMF, a DMT recommended for the treatment of MS, can reduce the amount but not the avidity of the anti-EBV humoral immune response in RRMS patients from the very early stages of treatment. Known under the trade name Tecfidera, DMF originated as a drug for the treatment of psoriasis, however, a 2012 study found that it could also be an effective treatment for MS [21]. Although the mechanisms by which DMF can reduce inflammation in MS are not yet fully understood, evidence suggests its potential antioxidant and neuroprotective role through the Nrf2 pathway [22]. The transcription factor Nrf2 plays a central role in the control of gene expression with antioxidant activity, thus exerting an important anti-inflammatory action [23].

Recent work by Bjornevik and colleagues demonstrated that EBV infection markedly increases the risk of subsequently developing MS, supporting its role in the pathogenesis of the disease [15]. Previous studies had also shown that there may be an association between specific antibodies to EBV antigens, especially EBNA-1 and CA, and some clinical features of MS, such as disease initiation and activity [6,9,10,11,12,13,24,25]. These antibodies could be considered putative biomarkers to describe the natural history of the disease, or “type 0 biomarkers” as defined by Bielekova and Martin [26]. We aimed to investigate whether EBV-specific antibodies could also be used in RRMS patients as “type I biomarkers” to capture the effects of DMF therapeutic intervention through possible fluctuations of serum levels of these antibodies over time. Humoral immunity against EBV is likely to be maintained constant for life because, in contrast to acute viral infections, chronic and latent viral infections can persist and be reactivated by latency, keeping immune responses stable during the life of the infected person [27]. In our study, serum levels of anti-CA IgG decreased immediately in the first seven days of treatment, and this reduction was stable over time for the first six months of therapy, suggesting that the decrease in antibody levels may be a direct consequence of DMF treatment. It is interesting to note that this reduction is appreciable both by comparing the median values at the various time points and as a trend in the concentration variations, expressed as deltas, with respect to the pre-treatment. This ability of DMF treatment to affect serum EBV-specific antibody levels from the first days of therapy was also partially confirmed by serum fluctuations in EBNA-1-specific IgG. EBV remains the environmental factor mainly associated with the development of MS, and to the best of our knowledge, this is the first time that EBV-specific antibody titres have been able to capture the effects of a disease-modifying treatment in MS patients so early. Recently, it has also been demonstrated that Natalizumab (Tysabri, Biogen Idec Inc, Cambridge, MA, USA), a humanized anti–α4 integrin monoclonal antibody considered a highly effective treatment for RRMS, is capable of decreasing serum levels of anti-EBV glycoprotein 350 IgG after 12 months of therapy, whereas a previous interferon beta treatment maintained those relatively stable [16]. Previous studies failed to find this decline in anti-EBV antibody levels [28,29], or on the contrary found only a slight and reversible increase in anti-EBV CA IgG, during natalizumab treatment [30]. As highlighted by the authors themselves, these discrepancies could be the result of the different sizes of the samples analysed in the respective studies or, above all, of the different choices of the target antigen used (EBV glycoprotein 350 vs. EBNA-1 and CA). In our samples, mainly anti-CA IgG and less so anti-EBNA-1 IgG were decreased. As CAs are viral surface proteins and EBNA-1 represents nuclear viral proteins, our result may be a consequence of the different biological significance of these two different antibodies. It has been postulated that EBV acts as an intermittently cytopathic virus [31] that can latently persist for life in B-cells, inducing recurring reactivations [32]. Lytic proteins, including CA, are expressed during replication, whereas latent genes, including EBNA, are expressed in the growth phases of infection [6]. The decrease we found in the serum anti-CA IgG levels immediately after the first days of therapy may reflect a reduction in replicative EBV infection [31,32]. Beyond the attempt to explain our results with what is already known about EBV and MS, our study has no mechanistic ambitions, and therefore only future studies will be able to clarify the actual effect of DMF on the control of EBV-specific humoral immunity in individuals with SM. High levels of specific antibodies to EBV capsid antigen have previously been associated with increased disability and worsening of disease detectable on magnetic resonance imaging [33]. In our study, we found no correlation between serum anti-EBV IgG levels and disease disability, expressed by the EDSS scale. Moreover, for the whole duration of our study, no relapses or increases in disability were observed. Only studies with longer follow-ups will be able to clarify whether or not there is a correlation between a reduction in the anti-EBV antibody response and better clinical conditions of the patients.

It is important to underline how our study analysed the avidity of EBV-specific antibodies for the first time longitudinally and in response to a DMT. To determine with certainty an ongoing infection, for some years researchers have begun to analyse the avidity of IgG antibodies; in fact, the first response of the immune system to infection is the formation of low-avidity antibodies [17]. As the infection progresses, antigen-specific IgG is formed and avidity increases. In this way, if high-avidity IgG is not yet present in the serum, it is possible to consider the infection still in its initial stage. To identify low-avidity antibodies in patient serum, two ELISA tests were performed in parallel: one test was performed conventionally, while the other involved the use of a urea solution, which was added to the wells of the microplate, between the incubation of the serum samples and the incubation of the enzyme conjugate. Urea caused the cleavage of low-avidity antibodies bound to the antigens and allowed only high-avidity antibody determination [34]. All of our patients had high-avidity anti-EBV CA antibodies before initiation of treatment, indicating that EBV infection had occurred in the past in all subjects enrolled in the study. Furthermore, although DMF therapy was able to reduce the amount of CA-specific antibody response, the avidity for the target antigen was not changed, suggesting that treatment may somehow reduce EBV antigenic stimulation without affecting the differentiation status of antibody-secreting cells.

The main limit of this study is the lack of the total serum IgG analysis in parallel to the EBV-specific IgG response, because the decrease we observed could be considered a consequence of general immune suppression. However, DMF is an immunomodulatory rather than an immunosuppressive treatment and, although it may be associated with lymphopenia, there is no evidence supporting a decrease in serum IgG levels during the treatment. Moreover, serum IgG measurement is not even recommended as a screening test before starting DMF or during treatment as a follow-up test (https://www.ema.europa.eu/en/documents/product-information/tecfidera-epar-product-information_en.pdf, (accessed on 10 January 2023)). In support of this, Longbrake and collaborators demonstrated that the humoral immunity is not affected by DMF, and total IgM, IgA, IgG and IgG1-4 subclass levels remain stable over 2 years of DMF treatment [35]. Literature evidence also supports that DMF does not affect the immune response to vaccines [36,37]. Considering overall this evidence, we could not *a priori* expect a decrease in EBV antibodies during DMF treatment nor can we explain it with a total IgG level decrease. Furthermore, our study found that DMF treatment does not affect different EBV-specific serum IgG to the same extent, as it appears to affect anti-CA IgG more than anti-EBNA-1 IgG.

Other limits of our study were certainly the small sample size and the short follow-up period, however, the homogeneity of the population and the accuracy and frequency of the different time points taken together should give consistency to our results.

In conclusion, we demonstrated for the first time that oral treatment with DMF can quantitatively, but not qualitatively, reduce the EBV-specific antibody response in MS patients from the very first days of therapy. In particular, the serum levels of specific IgG for capsid antigen could therefore be considered a potential biomarker for monitoring the therapeutic response to DMF in patients affected by RRMS. However, future studies are needed to investigate the effective role of the EBV-specific antibody response in MS subjects treated with DMF.

## 4. Materials and Methods

### 4.1. Study Design, Population and Sampling

Patients were consecutively recruited in a longitudinal prospective study at the MS Centre, University Hospital of Ferrara, northern Italy, and sampled between January 2018 and January 2020. Inclusion criteria were the (i) diagnosis of RRMS according to the 2010 McDonald criteria or 2017 revised McDonald criteria [4,38], (ii) age between 18 and 65 years old and (iii) being potential candidates to start treatment with DMF according to good clinical practice. Exclusion criteria were current DMF treatment, antibiotic treatment or high-dose corticosteroids within 30 days before study enrolment, current pregnancy and current or recent (within 12 months before enrolment) treatment with immunosuppressants.

Seventeen RRMS patients were included in the study. A starting DMF dose of 120 mg orally twice a day for 7 days followed by the maintenance dose of 240 mg orally twice a day was prescribed for each patient.

Information about MS history was recorded for each subject, in particular the year of onset and disease severity, scored using Kurtzke’s EDSS [20]. Neurological examination and blood collection were performed before starting DMF therapy (T0), and after 1 week (T1) and 1 (T2), 3 (T3) and 6 (T4) months of treatment. All the clinical data were collected by a neurologist with expertise in MS (C.F., M.L., E.B., M.P.).

The approval of the Ethics Committee of Area Vasta Emilia Centrale (AVEC), Italy, was obtained for experiments involving human subjects as well as written informed consent from all subjects participating in the study (protocol code 170288, 14 September 2017). Serum was obtained after blood centrifugation at 2000 g at 20 °C for 15 min, collected under sterile conditions in aliquots of 500 µL, coded, and stored at −80 °C until assay. All samples were analysed under the same conditions.

### 4.2. Serum Levels of Anti-Epstein–Barr virus Antibodies

In all serum samples, concentrations of anti-EBNA-1 and anti-CA IgG were measured by ELISA using commercially available kits (Euroimmun, Lübeck, Germany, Anti-EBNA-1 and anti-CA ELISA IgG, order numbers EI 2793-9601 G and EI 2791-9601 G, respectively) following the manufacturer’s instructions. Both products were marked as “in vitro diagnostic” (IVD) devices and carried the “CE” mark indicating compliance with the current European directive on in vitro diagnostic devices. All reagents, plates and peroxidase-conjugated antibodies were included in the kit. Microtiter strip wells were precoated with EBNA-1 and CA, respectively. Briefly, 100 µL of serum samples, prediluted 1:202, were dispensed in duplicate into two microtiter plates, one precoated with EBNA-1 and the other precoated with CA. After 30 min of incubation at room temperature (RT) and three washing cycles, 100 µL of a peroxidase-conjugated antibody specific for human IgG was added to each well and incubated for a further 30 min at RT. After three washing cycles, 100 µL/well of chromogen/substrate solution were applied and the plate was incubated for 15 min at RT protected from direct light. Finally, 100 µL of stop solution was added to each well. A photometric measurement was conducted at 450 nm using a second reference wavelength of 630 nm as control. EBNA-1 and CA-specific IgG levels were expressed as optical density (OD) in all samples, according to a previous study [16]. To avoid inter-assay variability, all samples from the same patient (T0, T1, T2, T3 and T4) were measured in the same assay run.

### 4.3. Avidity Determination of Anti-Epstein–Barr Virus Antibodies

To quantify antibody avidity in patient serum, two ELISA tests were run in parallel: one test was performed conventionally, as described in the previous paragraph, while the other involved the use of a solution of urea, which was added to the wells of the microplate, between the incubation of serum samples and incubation of the enzyme conjugate (anti-human IgG antibodies conjugated with peroxidase). The urea caused the detachment of low-avidity antibodies bound to the antigens and therefore allowed the investigation of only high-avidity antibodies. Serum avidity distributions of anti-CA IgG were measured using a commercially available kit (Euroimmun, Anti-EBV-CA ELISA, order number EI 2791-9601-1 G). The product was marked as an ‘in vitro diagnostic’ (IVD) device and carried the: CE” mark indicating compliance with the current European directive on in vitro diagnostic devices. All reagents, plates and peroxidase-conjugated antibodies were included in the kit. Only serum samples collected at baseline (T0) and consecutively at 1st (T2), 3rd (T3) and 6th months (T4) after the initiation of DMF treatment were analysed. Briefly, 100 µL of each serum sample (dilutes 1:101) was added to two adjacent wells of a precoated ELISA microplate. After 30 min of incubation at RT and one washing cycle with 300 µL of wash buffer, 200 µL of urea solution or 200 µL of phosphate buffer was applied into the wells corresponding to each patient for 10 min at RT. After three washing cycles, 100 µL of peroxidase-labelled anti-human IgG was dispensed into each well. After 30 min of incubation at RT and three washing cycles, 100 µL of chromogen/substrate solution was applied to each well. After 15 min of incubation at RT protected from direct light, 100 µL of stop solution was added to each well. Finally, a photometric measurement was made at a wavelength of 450 nm using a second reference wavelength of 630 nm as control. The RAI, expressed as a percentage, was calculated through the ratio between the OD of the sample treated with urea multiplied by 100 and the OD of the same sample without urea treatment. RAI values were analysed quantitatively as a percentage value and compared at any time points or interpreted qualitatively as indicated by the manufacturer: RAI < 40%, low-avidity antibodies; RAI 40%–60%, equivocal avidity; RAI > 60%, high-avidity antibodies. As for the previous methods, all samples from the same patient (T0, T2, T3 and T4) were measured in the same assay run to avoid inter-assay variability.

### 4.4. Delta Values

We investigated changes in anti-EBNA-1 and anti-CA IgG levels, as well as in anti-CA IgG RAI, in all patients during DMF treatment. The OD and the RAI value of the first sample taken at T0 (pre-therapy) were subtracted from the values of the subsequent time points: T1 minus T0, T2 minus T0, T3 minus T0 and T4 minus T0 for the IgG levels; T2 minus T0, T3 minus T0 and T4 minus T0 for RAI values. Delta (∆) OD and ∆RAI values were calculated accordingly. Values above zero were considered suggestive of an increase in anti-EBV antibody levels or antibody avidity, respectively, while values below zero indicated a reduction in anti-EBV antibody levels or avidity.

### 4.5. Statistical Analysis

Due to the small size of our cohort and the lack of normality in many of the data distributions, checked with the Kolmogorov–Smirnov test, a non-parametric statistical approach was used. Continuous variables were reported as the median and interquartile range (IQR). All tests were summarized in Table 3. Briefly, the Friedman test and Dunn’s post hoc test for repeated measures were used to compare anti-EBV IgG levels and RAI values. The ∆OD and ∆RAI values were compared with the Wilcoxon Signed Rank Test at all time points. The association between serum levels of anti-EBV IgG and the disease severity, expressed through the EDSS, was investigated with the Spearman’s rank correlation. A two-tailed *p*-value < 0.05 was considered statistically significant. Prism 9 for MacOS (GraphPad Software, La Jolla, CA, USA) was used for the statistical analysis.

## Figures and Tables

**Figure 1 ijms-24-01500-f001:**
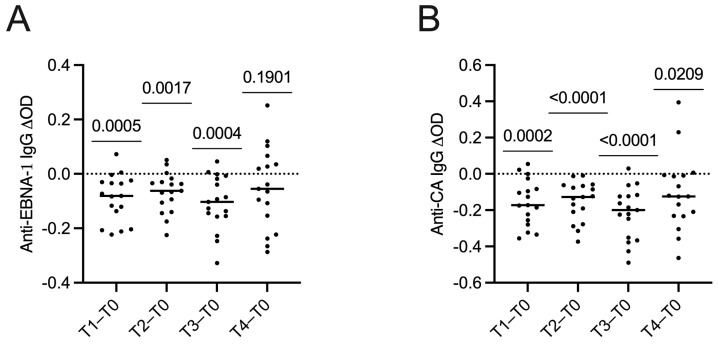
Changes in serum levels of anti-Epstein–Barr virus nuclear antigen-1 (EBNA-1) (**A**) and anti-capsid antigen (CA) (**B**) IgG expressed as delta (∆) optical density (OD) values in 17 relapsing–remitting multiple sclerosis patients during dimethyl fumarate treatment. Time points refer to pre-treatment (T0) and after 7 days (T1) and 1 (T2), 3 (T3) and 6 (T4) months of therapy. For each patient, the OD value of the first sample taken at T0 was subtracted from the further time points, calculating ΔOD values (T1−T0, T2−T0, T3−T0 and T4−T0). The ∆OD values were compared at all time points with the Wilcoxon Signed Rank Test.

**Figure 2 ijms-24-01500-f002:**
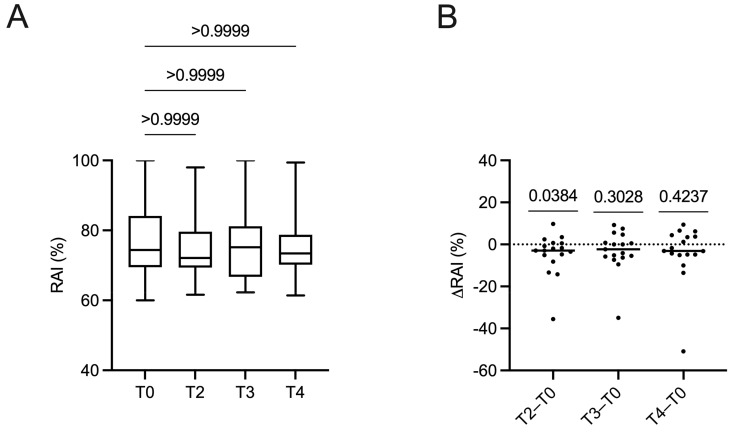
Avidity of anti-Epstein–Barr virus capsid antigen (CA) IgG expressed as relative affinity index (RAI) in 17 relapsing–remitting multiple sclerosis patients treated with dimethyl fumarate. Time points refer to pre-treatment (T0) and after 1 (T2), 3 (T3) and 6 (T4) months of therapy. Urea was used to measure antibody avidity. For each patient, the RAI, expressed as a percentage, was calculated through the ratio between the optical density (OD) of the sample treated with urea, multiplied by 100, and the OD of the same sample without urea treatment. The Friedman test and Dunn’s post hoc test for repeated measures were used to compare RAI at all time points (Panel **A**). The RAI value of the first sample taken at T0 was subtracted from the further time points, calculating ΔRAI values (T2−T0, T3−T0 and T4−T0). The ∆RAI values were compared at all time points with the Wilcoxon Signed Rank Test (Panel **B**).

**Table 1 ijms-24-01500-t001:** Demographic and clinical characteristics of 17 relapsing–remitting multiple sclerosis patients receiving dimethyl fumarate. EDSS, expanded disability status score; IQR: interquartile range.

	N = 17
Age at entry, years: median (IQR)	40.0 (31.0–45.5)
Women: n (%)	12 (70.6)
Disease duration, months: median (IQR)	29.0 (14.0–103.5)
EDSS score: median (IQR)	1.5 (1.0–1.5)

**Table 2 ijms-24-01500-t002:** Longitudinal fluctuations in serum levels of anti-Epstein–Barr virus nuclear antigen-1 (EBNA-1) and anti-capsid antigen (CA) IgG expressed as optical density (OD) in 17 relapsing–remitting multiple sclerosis patients during six months of dimethyl fumarate treatment.

	Time Points				
	T0 (Pre-Therapy)	T1 (1 Week)	T2 (1 Month)	T3 (3 Months)	T4 (6 Months)
Anti-EBNA-1 IgG levels, OD ^a^: median (IQR)	1.261 (1.061–1.477)	1.231 (0.889–1.444)	1.252 (0.966–1.423)	1.238 ^b^ (0.857–1.425)	1.221 (0.938–1.520)
Anti-CA IgG levels, OD ^c^: median (IQR)	1.308 (1.077–1.666)	1.121 ^d^ 0.932–1554)	1.186 ^e^ (0.942–1.514)	1.067 ^f^ (0.911–1.319)	1.182 ^g^ (0.954–1.601)

Median serum levels of ^a^ anti-EBNA-1 IgG and ^c^ anti-CA IgG were different between various time points (Friedman test, *p* = 0.0066 and *p* < 0.0001, respectively). The Friedman test and Dunn’s post hoc test for multiple comparisons were used to compare median OD values at all time points with the median OD value at T0. Anti-EBNA-1 IgG levels: ^b^ T3 vs. T0, *p* = 0.0014. Anti-CA IgG levels: ^d^ T1 vs. T0, *p* = 0.0136; ^e^ T2 vs. T0, *p* = 0.0014; ^f^ T3 vs. T0, *p* < 0.0001; ^g^ T4 vs. T0, *p* = 0.0009. IQR: interquartile range.

**Table 3 ijms-24-01500-t003:** Overview of statistical tests used in the manuscript. CA, Epstein–Barr virus (EBV) capsid antigen; EBNA-1, EBV nuclear antigen 1; EDSS, expanded disability status scale; OD, optical density; RAI, relative avidity index; ∆OD and ∆RAI, changes in OD or RAI values, respectively, between two different time points (e.g., T1 minus T0).

Analysis Type	Data Analysed (Units)	Test
Comparison between groups	Anti-EBNA-1 IgG levels (OD) Anti-CA IgG levels (OD) RAI (%)	Friedman test
Multiple comparison (each time point vs. every other time point)	Anti-EBNA-1 IgG levels (OD) Anti-CA IgG levels (OD) RAI (%)	Dunn’s pots hoc test
Changes in repeated measures	∆OD ∆RAI	Wilcoxon Signed Rank Test
Degree of association between variables	Anti-EBNA-1 and anti-CA IgG levels (OD) vs. EDSS score	Spearman’s rank

## Data Availability

The datasets used and analysed during the current study are available from the corresponding author on reasonable request.

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
