# Peer review of "Dimethyl Fumarate Treatment Reduces the Amount but Not the Avidity of the Epstein–Barr Virus Capsid-Antigen-Specific Antibody Response in Multiple Sclerosis: A Pilot Study"

_ijms, 2023, doi:10.3390/ijms24021500_

Round 1
Reviewer 1 Report
The study investigated the effects of DMF treatment on EBV-specific IgG in multiple sclerosis patients and revealed DMF reduced quantity not the quality of anti-EBV immune response. The manuscript is clearly written and easy to follow. A possible weakness is the small sample size and short follow-up time, as stated in the discussion section.
My specific concerns are:
1. This study focused on two antibodies, anti-EBNA-1 and anti-CA IgG. Are there any differences between them? More background information is needed in the introduction section.
2. When studying the changes in quantity and quality of anti-EBV IgG, the author compared different post-treatment time points with T0. It’s also interesting to investigate how these values changed during the post-treatment period. Did the authors observe any significant differences between OD or RAI at different post-treatment time points (e.g., T2 vs. T1)?
3. Figure 1A-B. It seems that the variance of ΔOD increased dramatically at T4. Could the author provide any interpretations? For some possibly exception dots far from the median value, did the author observe any interesting characteristics of these patients?
4. Line 115. Why was T1 not included for comparison for the avidity study?
5. A negative control group without DMF treatment would further consolidate the conclusions in this manuscript.
6. Some abbreviations emerged before or without full names. For example, line 67 “RRMS”, line 70 “EDSS”.
7. Line 114-121, Line 133-140. Section 2.4 is duplicated.
Reviewer 2 Report
In the current manuscript, entitled “Dimethyl fumarate treatment reduces the quantity but not the quality of the Epstein-Barr virus-specific antibody response in multiple sclerosis: a pilot study”, Castellazzi et al., investigated the relationship between DMF treatment and EBV infection. They showed that DMF treatment could reduce the EBV-specific IgG level which included anti-EBNA-1 IgG and anti-CA IgG. Further, they found that DMF didn’t reduce the avidity of anti-CA IgG at all time points.
In general, the authors provided interesting discovery that DMF treatment alleviates MS progression may through reducing the antigen level of EBV. The data is clear. The conclusion is supported by the data.
This manuscript is more suitable for publishing as report based on the current format of the manuscript.
I have 1 minor points.
Please define the concentration of urea solution in the method.
Reviewer 3 Report
This manuscript analyzed the serum anti-EBNA-1 and anti-capsid antigen IgG from 17 relapsing-remitting multiple sclerosis patients with ELISA. The ELISA was done with samples collected at five different time points (pre-therapy, 1 week, 1 month, 3 months, and 6 months after therapy). Compared with pre-therapy, the anti-EBNA-1 and anti-capsid antigen IgG amounts decreased after the therapy after different time points based on the OD change. The anti-CA IgG avidity didn’t obviously change after the therapy. Dimethyl fumarate is an immune suppressive, it’s no surprise that the dimethyl fumarate therapy will decrease the IgG amounts. Besides, there are also other questions that need to be solved.
1. The total IgG change is needed to be compared before vs after the therapy. If the total IgG also decreases after the therapy, the decrease of the anti-EBV antibodies could be just the consequence of the general immune suppression.
2. The method is not clear enough. Since the ELISA didn’t show the titer curve or EC50 nor a standard curve, only compared the OD, it’s hard to directly compare the OD change, especially among different ELISA plates.
3. Based on the manuscript, it assumed antibody avidity equal to antibody quality. How about antibody diversity? Antibody functions: neutralization, ADCC?
4. The avidity of anti-EBNA-1 is missing.
5. The manuscript only tested the antibody response for t=0 vs t= 1, 2, 3, and 4. Does the antibody response change inside t=1, 2, 3, and 4?
6. Is there any correlation between EDSS score with EBV antibody?
Reviewer 4 Report
This article by Castellazzi et al. performed a pilot study to show an effect of DMF treatment to reduce EBV specific antibodies in MS pathology. This is a simple observational study done in human patients, yet pointing towards a significant observation which might implicate therapeutics aspects of such drug treatments on MS patients. However, the whole study was done in a small sample size and for shorter period (no observation on long term treatment of the substance). But the authors have correctly pointed out these shortcomings and summed it well in their discussion section. Since the intent of this study seems to be on point with no extravagant claims, I think this can be published as a short observational pilot study. The article is well written and scientifically sound, and the observations are efficiently discussed with valid reasoning. I do not have any major comments.
Minor Comments:
1. Please define RAI when it is introduced for the first time in line 117.
2. Please provide a comprehensive table for individual statistical tests that were performed for different comparisons. This can be added into section 4.5, as an extra table, summarizing all the p-values at one place.
Round 2
Reviewer 3 Report
Thank you for answering my questions.
Author Response
We thank the Reviewer for the valuable suggestions and for the possibility that he/she has given us to improve our manuscript.
A check of the language and of possible typos has been carried out throughout the manuscript.